# Comparison of Antimicrobial Resistances in *Escherichia coli* from Conventionally and Organic Farmed Poultry from Germany

**DOI:** 10.3390/antibiotics11101282

**Published:** 2022-09-21

**Authors:** Mirjam Grobbel, Jens A. Hammerl, Katja Alt, Alexandra Irrgang, Annemarie Kaesbohrer, Bernd-Alois Tenhagen

**Affiliations:** 1Department Biological Safety, German Federal Institute for Risk Assessment, 10589 Berlin, Germany; 2Unit Veterinary Public Health and Epidemiology, University of Veterinary Medicine, 1210 Vienna, Austria

**Keywords:** antimicrobial resistance, broiler, turkey, organic farming, conventional farming

## Abstract

In this study, resistance rates in *Escherichia coli* from organic and conventional poultry in Germany were compared. Isolates were randomly collected from organic and conventional broiler and turkey flocks at the farm and from turkey meat at retail. Resistance testing was performed as prescribed by Commission implementing decision 2013/652/EU. Logistic regression analyses were performed for the resistance to the different antimicrobials. Overall, resistance rates for the antimicrobials tested were lower in *E. coli* from organic than from conventionally raised animals. In turkeys, the percentage of isolates susceptible to all antimicrobials tested from animals and meat was twice as high from organic than from conventional origin (~50% vs. <25%). In broilers, the percentage of susceptible isolates from organic farms was five times higher than from conventional farms (70.1% vs. 13.3%) and resistance to three or more classes of antimicrobials was 1.7- to 5.0-fold more common in isolates from conventional farms. The differences between organic and conventional farming were more pronounced in broilers than in turkeys. More studies on turkeys are needed to determine whether this difference is confirmed.

## 1. Introduction

Antimicrobial resistance is one of the major threats to human health and is responsible for the death of thousands of people every year, with an increasing trend [1]. One of the main drivers of antimicrobial resistance in bacteria is selection pressure through antimicrobial therapy or antimicrobial residues [2]. Most families of antimicrobial substances are used not only in human medicine but also in veterinary medicine. Livestock husbandry is one of the reservoirs of resistant bacteria considered relevant in the one health context [2]. Either directly via contact with animals or through the food chain, or indirectly through contamination of the environment, resistant bacteria can be transmitted from livestock to humans. To minimize this transmission, many countries have significantly reduced the use of antimicrobials in veterinary medicine in recent years [3]. As a result, the prevalence of resistant bacteria in livestock has partially decreased [4].

In Germany, the consumption of organic food is increasing, and with it the number of organic farms [5]. Organic animal husbandry must meet requirements that minimize negative effects on animal welfare and health [6]. The use of drugs, including antimicrobials is strictly regulated [7]. This suggests that resistant bacteria are less common in organically raised animals than in conventionally raised animals. Several studies address this issue, but most of them do not provide data representative of a country. In addition, for poultry, there are studies on broilers, but very few studies on turkeys [6,8,9,10]. Therefore, the objective of this study was, to look at the difference in the resistance between indicator *Escherichia coli* from broilers and turkeys from organic and conventional husbandry, collected as a part of the German national monitoring for zoonotic agents.

## 2. Results

A total of 332 *E. coli* from fecal samples of broilers at the farm level were included in this study. Of these 301 were from conventional, and 31 from organic husbandry. Furthermore, a total of 230 fecal *E. coli* from turkeys at the farm level (200 from conventional, 30 from organic) and 330 *E. coli* from turkey meat at retail were included (196 from conventional, 134 from organic).

Resistance to antimicrobials in isolates from different origins is given in Table 1, and an overview of the number of resistances per isolate is in Figure 1. Overall, the highest rates of resistant *E. coli* were found to ampicillin in all populations considered. In conventional turkeys and turkey meat, tetracycline resistance was second. In conventional broilers and organic turkeys, resistance to sulfamethoxazole was second. Interestingly, in isolates from organic broilers, resistance to quinolones (nalidixic acid followed by ciprofloxacin) was as frequent as resistance to tetracyclines and sulfonamides, sharing the second position in the ranking. In the other populations, resistance to quinolones (ciprofloxacin and/or nalidixic acid) was among the most frequent resistances observed. Resistance to gentamicin, cefotaxime, ceftazidime, colistin, and azithromycin was low (below 10%) but with some differences between origins. Resistance to meropenem was not observed and resistance to tigecycline was observed in one isolate only. For colistin, no resistant isolate was found from broilers or turkeys from organic farming, while 5.2% of the isolates from organic turkey meat were resistant. 

The probability of being fully susceptible was more than five times higher in *E. coli* isolates from organic farming than in isolates from conventional husbandry (OR 0.191 [CI 0.132; 0.277], Appendix A). In *E. coli* isolates from organic broiler farms this probability was almost 16 times higher than in those from conventional broiler farms (OR 0.063 [CI 0.027; 0.146], Appendix A). In isolates from organic turkeys, the probability of being fully susceptible was 3.5 times higher than in those from conventional turkeys (OR 0.284 [CI 0.129; 0.623], Appendix A). In isolates from organic turkey meat, it was 4.1 times higher compared to conventionally produced meat (OR 0.249 [CI 0.154; 0.404], Appendix A).

Resistance to three or more substance classes per isolate was significantly more probable in *E. coli* from conventional, compared to those from organic husbandry (OR 4.1, Appendix A). This difference was not significant in isolates from turkey at the farm level, but it was in those from turkey meat (OR 4.148 [CI 2.524; 7.095]) and broiler farms (OR 8.675 [CI 2.425; 7.095], Appendix A). 

Regarding the individual substances, the difference was observed in all three origins (i.e., broilers and turkeys at farm and turkey meat) for ampicillin and tetracycline. For a number of substances (sulfamethoxazole, trimethoprim, nalidixic acid, and ciprofloxacin), the difference was observed in the overarching analysis (Appendix A), for broilers on the farm and in turkey meat at retail but not observed in isolates from turkeys at the farm (Appendix A). 

The difference in the resistance to chloramphenicol between *E. coli* from organic and conventional farming was significant in the overarching analysis (Appendix A) but not in the individual categories. For colistin, even though significant in the overarching analysis, the difference in the resistance percentage between isolates from organic and conventional farming was not significant for any of the origins when analyzed separately.

Resistance to gentamicin, azithromycin, and the third-generation cephalosporins cefotaxime and ceftazidime was overall low with resistance rates of less than 8% in all six sample categories. Resistance to gentamicin was more frequent in turkeys and turkey meat than in broilers but did not differ between organic and conventional production. No isolates from organic broiler farms were resistant to cefotaxime or ceftazidime. However, in two isolates from organic turkey farms (6.7%) and 3.0% of isolates from organic turkey meat, resistance to one or both of the substances was found. Resistance to the third generation cephalosporins did not differ significantly between the three origins nor between organic and conventional production.

## 3. Materials and Methods

### 3.1. Study Design

The study was carried out in the framework of a national monitoring program for Germany in 2016 and 2018 set up by the authorities of the federal states, the Federal Office for Consumer Protection and Food Safety (BVL), and the German Federal Institute for Risk Assessment (BfR) to fulfill requirements of Directive 2003/99/EC as well as more specific requirements set in CID 2013/652/EU [11]. The national framework of the monitoring is regulated by the “General Administrative Regulation on the Collection, Evaluation and Publication of Data on the Occurrence of Zoonoses and Zoonotic Agents in the Food Chain (AVV Zoonosen Lebensmittelkette)” [12]. 

In 2016 *E. coli* from broiler fecal samples at farm and in 2018 *E. coli* from turkeys at farm and turkey meat at retail were investigated, each originating from both production types [13,14]. Samples from conventional and organic farms were collected all over Germany. Conventional farms were selected in each federal state proportionate to the number of broilers or turkeys housed in the respective federal state. Due to the low number of organic farms, all farms of this production system that housed more than 1000 broilers or 250 turkeys were included in the sampling frame. At farms, one pair of boot swabs per flock was collected.

Sampling of meat at retail was distributed all over Germany, proportional to the number of inhabitants in the federal state. All sampling had to be performed by official veterinarians. 

### 3.2. Isolates

Indicator *E. coli* were isolated from broiler and turkey fecal samples at farm level (boot swabs), and turkey meat, at the accredited regional laboratories. Isolates were sent to the National Reference Laboratory for Antimicrobial Resistance (NRL-AR) at the BfR.

At BfR, the isolates were cultured on ENDO-Agar (ThermoScientific, Meerbusch; Germany). If the colonies did not show typical *E. coli* morphotype, species was confirmed using MALDI ToF (Maldi BioTyper microflex LT/SH, Bruker, Bremen, Germany). Isolates were stored for further analyses at −80 °C in Lysogeny Broth supplemented with 40% Glycerol.

### 3.3. Antimicrobial Susceptibility Testing (AST)

*E. coli* isolates were tested for their susceptibility to a fixed panel of antimicrobials using the broth microdilution method, following CLSI guidelines M07-A9 [15]. Commercial microtitre plates (Sensititre^®^, ThermoScientific, Meerbusch, Germany) with the EUVSEC layout were used. They contained the 14 antimicrobial substances of 12 antimicrobial classes ampicillin, azithromycin, cefotaxime, ceftazidime, chloramphenicol, ciprofloxacin, colistin, gentamicin, meropenem, nalidixic acid, sulfamethoxazole, tetracycline, tigecycline, and trimethoprim, in concentration ranges described by CID 2013/652/EU [11]. Minimum inhibitory concentrations (MIC) were evaluated according to the epidemiological cut-off values (ECOFF) from the European Committee for Antimicrobial Susceptibility Testing (EUCAST) and tentative ECOFFS from the European Food Safety Authority (EFSA), as laid down in CID 2013/652/EU [11,16]. Isolates with MIC-values up to the ECOFF were considered “wildtype” and are further classified as “susceptible”. Results above the ECOFF were considered “non-wildtype”, presumably carrying some resistance mechanism, and are further classified as “resistant”. Isolates, which showed no resistance to any of the tested antimicrobials are further classified as “fully susceptible”, isolates with resistance to three or more of the tested classes of antimicrobials as “multiresistant”.

### 3.4. Statistical Analyses

Data were analyzed by binary logistic regression analyses using IBM^®^ SPSS^®^ software version 26. Resistance of *E. coli* to the individual antimicrobials and the categories “fully susceptible” and “multiresistant” were the binary outcome variables. Production type (organic vs. conventional and source (broiler fecal samples, turkey fecal samples, and turkey meat were the independent variables) (Appendix A). The categories “broiler at farm” and “organic” were taken as reference. 

For a second set of logistic regression analyses data were analyzed separately by source (i.e., “broiler at farm”, “turkey at farm” and “turkey meat” and only type of production was considered as independent variable (Appendix A). 

Level of significance was set at alpha = 0.05.

The farming system was coded with 0 = organic and 1 = conventional.

## 4. Discussion

We compared resistance data of *E. coli* from conventional and organic broilers and turkeys, as well as from conventional and organic turkey meat at retail. The samples were collected in the framework of national monitoring [13,14]. The results provide a picture of the resistance situation of a representative set of *E. coli* from German poultry production. Although there are studies comparing resistance data of isolates from organic and conventional poultry production, only a few use data from national monitoring programs collecting representative data for an entire country [10].

*E. coli* isolates from organic production were more likely to be fully susceptible than those from conventional production. This is in line with our hypothesis and for broilers, it is in line with recent studies from Italy and Austria comparing conventional and organic broiler production [17,18]. For turkey farms, this has previously been observed for *Campylobacter* in the US [19] and Germany in a smaller study [8]. Isolates of *Campylobacter* jejuni from organic turkey meat were likewise more likely to be fully susceptible than isolates from conventional meat [10]. Data for comparison of *E. coli* isolates from organic and conventional turkey farms is scarce. Mughini-Gras et al. compared antimicrobial-resistant bacteria and immune markers of turkeys from both husbandry forms [6]. They also found lower numbers of resistant isolates in organic than in conventional husbandry but were not able to show significant differences. However, they only investigated four *E. coli* isolates from organic and 28 from conventional farms and therefore lacked the statistical power to demonstrate differences.

Multiresistance, on the other hand, was more frequently observed in conventional than in organic production (Appendix A). This was confirmed in the individual models for broilers at farm and turkey meat at retail. However, the difference was not significant for turkeys on the farm despite an OR of 2.283. The latter points to the limitations of power in the two on-farm studies as only a few organic farms could be included. 

Looking at the individual substances, the differences between conventional and organic production were significant for all three origins of ampicillin and tetracycline. Differences in AMR between organic and conventional production have previously been reported in broilers by other studies [17,18]. However, a study from the US did not find differences between isolates from conventional and organic broiler meat, but for resistance to ampicillin between isolates from conventional and organic turkey meat [20]. In that study, the level of resistance in broiler meat was substantially lower than the level observed in turkey meat. Likewise, in a study on *Campylobacter* collected in the USA in 2012, differences between organic and conventional production were more pronounced in turkey than in broiler production and the level of resistance was substantially higher in isolates from turkeys [19]. These substantial differences between broilers and turkeys were not observed in our study. 

Other available data for the comparison of conventionally and organically raised turkeys mainly concern *Campylobacter* species. Two studies also include tetracycline resistance rates in turkeys on the farm [19] and turkey meat [10] from both farming systems, and in both resistance was significantly lower in isolates from organic than in those from conventional husbandry.

Failure to detect differences between conventional and organic turkeys is probably a consequence of a lack of statistical power to detect differences that were caused by the limited number of organic isolates from turkey flocks. In line with that, numerically, resistance to all four substances was higher in isolates from conventional turkeys than from organic turkeys. In the future, this comparison should be repeated including more organic turkey farms.

No regression analysis could be performed for some substances. Resistance to meropenem was absent in all isolates and resistance to tigecycline was only observed in one isolate. In line with that, resistance to meropenem is extremely rare in isolates of *E. coli* from broiler and turkeys in Europe and likewise, resistance to tigecycline is the exception [4].

No difference was observed for the third-generation cephalosporins cefotaxime and ceftazidime either. These substances are classified by WHO as the highest priority critically important antimicrobials [21] and therefore should not be used in animal production. Accordingly, they are not approved for use in poultry in Europe, so differences between organic and conventional production cannot be associated with differences in use. While resistance to cefotaxime has decreased over time in broilers in Germany, it has been consistently low in turkeys [4]. A recent study from Italy likewise did not find a significant difference in cefotaxime resistance between isolates from conventional and organic broilers [17].

Using selective cultivation of third-generation cephalosporin-resistant *E. coli* performed in parallel during the national monitoring showed quite a high proportion of resistant isolates from the same samples [13,14]. For samples from conventional broiler and turkey farms and turkey meat, the percentage of resistant *E. coli* was 50.2%, 51.8%, and 37.6%, respectively. From organic production, it was in general lower with 25.7%, 36.8%, and 12.2%, respectively.

Chloramphenicol is also not used in poultry, as it has been banned from use in food animals in Europe about 30 years ago. Likewise, in Germany florfenicol is not approved for use in poultry. Compared to other non-licensed substances resistance to chloramphenicol is still comparatively high in isolates from broilers and turkeys in many countries [4]. In contrast to the cephalosporins, resistance to chloramphenicol was overall less frequent in organic production than in conventional production (Appendix A), which is in line with the results of the Italian study on organic and conventional broilers [17]. However, the difference was not confirmed in the individual analyses of the three origins (Appendix A). Resistance to chloramphenicol has decreased over time after the ban, but due to co-selection with resistance to other antimicrobials, it is still prevalent in the population. Due to co-selection, the use of other substances might foster resistance to chloramphenicol. In line with that, it has been shown that resistance to chloramphenicol was associated with the overall level of antimicrobial use [22]. Unfortunately, stratified data on antimicrobial use are not available for organic and conventional poultry in Germany, i.e., this association cannot be studied in the framework of our study.

Resistance to gentamicin and azithromycin was low in all populations and this is in line with the low use of gentamicin in poultry production [23]. Azithromycin is not licensed for use in poultry. Therefore, differences in use were not a likely cause of differences in resistance to these. No difference in resistance to gentamicin between conventional and organic turkey meat was also observed for *Campylobacter* in a recent study carried out in the same framework [10]. In the US, the resistance of *E. coli* to gentamicin was more frequent in *E. coli* from conventional than in organic broiler meat, which was attributed to the use of gentamicin in ovo in hatcheries [20]. Likewise, in an Italian study, resistance to gentamicin was less frequent in isolates from organic broilers as compared to conventional ones [17]. 

Resistance to colistin differed significantly between organic and conventional production (Appendix A). However, there were no resistant isolates in organic farms and therefore it was not possible to run the statistical model validly for broilers and turkeys on farm. The failure to detect colistin-resistant isolates in organic farms could be associated with the limited sample size and should not be overestimated. However, it is in line with results from the Italian study that also did not find colistin-resistant *E. coli* in organic broilers [17]. Whether the colistin-resistant isolates in turkey meat originate from primary production or are a consequence of cross-contamination at slaughter cannot be defined. It has been pointed out that resistance to colistin overall is comparatively high in poultry in Germany [24] which is related to the high use of this substance in broilers and turkeys [22]. 

Our study has a few limitations: Due to the low market share of organic poultry farms, the exclusion of very small herds, and the decision to only test one isolate per epidemiological unit, the number of isolates from organic broilers and turkeys was limited which reduces the statistical power to detect differences between the populations. This could not be overcome and therefore this study should be repeated in the future and in other countries to confirm the results. In contrast, a sufficient number of isolates from food was available. The purpose here was to collect isolates that were representative of the exposure of consumers through food. 

Testing only one randomly selected isolate also implies that within the respective population there might be other *E. coli* with different characteristics. It was not the purpose of this work to characterize the AMR situation within the population in a detailed manner but rather to see whether there were differences between the populations on a large scale. 

We only analyzed phenotypic resistance and did not include molecular analyses, hence the potential dominance of certain *E. coli* types in one population or the other could not be discerned. 

## 5. Conclusions

This is the analysis of representative resistance results of *E. coli* from conventional and organic poultry production in Germany. Our results confirm our hypothesis that resistance rates are overall lower in organic than in conventional production. Differences were not observed for antimicrobials that had low resistance rates in all populations, most probably because they are either rarely used or not used at all. In some instances, a numeric difference was not confirmed by statistical testing, probably because of a lack of statistical power associated with too few isolates from organic farms. Future studies should on the one hand investigate which factors in animal husbandry are responsible for the observed differences. On the other hand, it should be investigated if isolates from organic and conventional production differ in the genetic background of resistance or aspects other than antimicrobial resistance such as virulence factors using molecular methods.

## Figures and Tables

**Figure 1 antibiotics-11-01282-f001:**
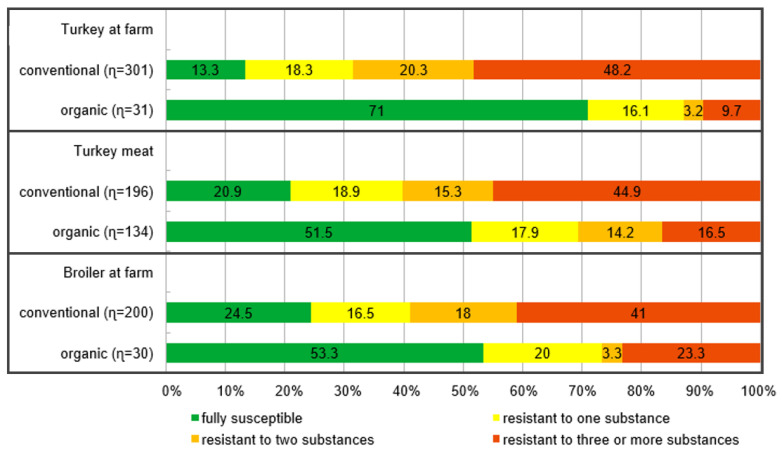
Percentage of isolates from the different categories fully susceptible, or resistant to one, two, or three or more classes of antimicrobial substances (*n*= number of isolates tested).

**Table 1 antibiotics-11-01282-t001:** Percentage of isolates resistant to an antimicrobial substance (conv = conventional farming, org = organic farming, GEN = gentamicin, CHL = chloramphenicol, FOT = cefotaxime, TAZ = ceftazidime, NAL = nalidixic acid, CIP = ciprofloxacin, AMP = ampicillin, COL = colistin, SMX =sulfamethoxazole, TMP = trimethoprim, TET = tetracycline, AZI = azithromycin, MERO = meropenem, TGC = tigecycline).

Category (Number of Isolates)			GEN	CHL	FOT	TAZ	NAL	CIP	AMP	COL	SMX	TMP	TET	AZI	MERO	TGC
**broiler**	**farm**	**conv** (*n* = 301)	1.3	7.6	1.7	1.7	41.5	44.5	70.4	8.3	59.1	52.5	40.2	1.7	0.0	0.0
		**org** (*n* = 31)	3.2	0.0	0.0	0.0	9.7	9.7	22.6	0.0	9.7	6.5	9.7	0.0	0.0	0.0
**turkey**		**conv** (*n* = 200)	5.0	20.0	1.5	1.5	23.0	35.0	65.5	9.0	36.0	23.0	49.0	2.5	0.0	0.5
		**org** (*n* = 30)	0.0	10.0	6.7	6.7	13.3	16.7	40.0	0.0	20.0	13.3	16.7	3.3	0.0	0.0
	**meat**	**conv** (*n* = 196)	7.7	21.9	2.6	1.5	23.5	35.7	63.8	6.6	41.8	32.7	56.6	1.5	0.0	0.0
		**org** (*n* = 134)	4.5	11.2	3.0	2.2	12.7	16.4	36.6	5.2	16.4	9.7	26.9	0.0	0.0	0.0

## Data Availability

The data presented in this study are available in the Appendix A.

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
