# Peer review of "Comparison of Antimicrobial Resistances in Escherichia coli from Conventionally and Organic Farmed Poultry from Germany"

_antibiotics, 2022, doi:10.3390/antibiotics11101282_

Round 1

Reviewer 1 Report

I have only some minor editorial remarks:

All species and genus names should be put in italics.

Check spelling of names in the References (namely, nos. 9 and 10)

Some references should be supplied with URL (website) or the issue and pages in official documents. This applies, in particular, to Refs. 11, 16, 17.

Ref. 7: At the time of the study, Regulation 834/2007 was in force but it was later replaced by Regulation 2018/848. I suggest to add this information to Ref 7.

Author Response

Dear Reviewer,

thank you for the positive feedback and your valuable comments to our manuscript.

  • We have now italicized all species and genera.
  • We have fixed the incorrect references.
  • We have added missing URLs and merged references 11 and 16.
  • We completely replaced the reference to Regulation 834/2007 by reference to 2018/848.

To help you track the changes, we are attaching the Word document with the change tracking of the improvements in the manuscript revised after the review.

Reviewer 2 Report

The manuscript deals with an interesting topic, it is well written and the English language is very good. The Authors present the results of a nation-wide investigation on Escherichia coli resistance in conventional and organic broilers and turkeys in Germany. Even though the topic is well-known, the Authors add new information and this is useful to better understand the issue. Furthermore, the methods adopted are appropriate and clearly outlined, as well as the results and the discussion are clearly presented. Conclusions are supported by the data presented.

I have just a few minor suggestions:

Lines 79-83: should be moved to the Results section

Tables 2 and 3 are very hard to read. If not possible to change format for presenting data, I suggest to include them as supplementary material.

Author Response

Dear Reviewer,

thank you for the positive feedback and your valuable comments to our manuscript.

  • We have moved lines 79-83 to the Results section (Lines 111-115 in the revised pdf-version).
  • We have removed Tables 2 and 3 from the manuscript and included them as supplementary material.

To help you track the changes, we are attaching the Word document with the change tracking of the improvements in the manuscript revised after the review.
